



# Is the North West European Shelf becoming more stratified with the occurrence of marine heatwaves?

Wei Chen[1], Joanna Staneva[1]

[1] Institute of Coastal Systems-Analysis and Modeling, Helmholtz-Zentrum Hereon, Max-Planck-Straße 1, Geesthacht, 21502,
Germany

*Correspondence to*: Wei Chen (wei.chen@hereon.de)

**Abstract.** Marine heatwaves (MHWs) are characterized by anomalous and prolonged increases in sea surface temperatures, driven by atmospheric and oceanic factors. The intensification of MHWs is an evident consequence of ongoing global climate change. The question of whether the North West European Shelf (NWES) is experiencing increased stratification in recent

decades is of significant interest in understanding the impacts of these extreme events. In this study, we leverage ocean physics reanalysis data obtained from Copernicus Marine covering the temporal span from 1993 to 2023 to conduct a rigorous examination of the NWES domain. The focus centers on the assessment of potential energy anomaly (PEA) and its role in shaping stratification dynamics.

Our findings reveal an increase in both the frequency and duration of MHWs in the NWES area, especially in coastal areas

where the duration of MHWs is increasing the fastest, generally by more than 2 days per year over the study period. However, despite the intensified MHWs, thermal stratification in the NWES is weakening, particularly in the middle and northern North Sea. This suggests that the warming effect due to MHWs is insufficient to counteract the overall decline in thermal stratification caused by global warming. Additionally, our study highlights the significance of seawater salinity in driving the trend of density stratification. Specifically, the discharge from the Baltic Sea plays a crucial role in influencing the stratification patterns

in the North Sea region. The outcomes of this research transcend theoretical confines, bearing practical significance for diverse sectors. By unravelling the intricate interplay between MHWs, thermal stratification, and salinity dynamics, our study contributes to a more comprehensive understanding of climate change impacts on regional oceanic systems. The implications extend to domains such as ecosystem dynamics, fisheries, and related sectors, which are poised to be influenced by the enduring alterations in thermal stratification patterns that have far-reaching implications for the ecological and socio-economic fabric

of the NWES region.

## 1 Introduction

Marine heatwaves (MHWs) are extreme oceanic events characterized by unusually warm sea surface temperatures (SSTs) that exceed the local 90th percentile for at least five consecutive days (Hobday et al., 2016). These events are characterized by their intensity, duration, and spatial extent, often leading to ecological disturbances and significant shifts in species distribution





patterns (Frölicher et al., 2018). MHWs can occur in various oceanic regions, including coastal areas, and the occurrence of MHW events has shown an increasing trend globally over the past century (Oliver et al., 2018, IPCC, 2021).

Climate models project a continued upward trend in the occurrence of MHWs in the coming decades, driven by anthropogenic climate change (Frölicher et al., 2018; Oliver et al., 2019, 2020, IPCC, 2021). The North West European Shelf (NWES), which

is a large area of shallow temperate seas located between 47 ºN~61 ºN latitude and 12 ºW~10 ºE (Figure 1), is expected to experience a similar trend of increasing MHW events (IPCC, 2021). Understanding the dynamics of MHWs and their consequences is crucial for effective ecosystem management and conservation efforts (Smale et al., 2019).

Elevated SSTs can lead to widespread and severe ecological disturbances, including shifts in species distributions, alterations

in community structure, and increased vulnerability to invasive species (Oliver et al., 2018; Smale et al., 2019). These events can disrupt important ecological processes, such as nutrient cycling, primary production, and trophic interactions, with cascading effects on the entire marine food web (Wernberg et al., 2013, 2016; Oliver et al., 2020). In the NWES region, the increasing frequency of MHW occurrence is anticipated to have significant consequences, for both socioeconomic system and natural processes. For instance, Borges et al. (2019) observed a threefold increase in dissolved methane concentration in surface

waters along the Belgium coast during summer of 2018 compared to a typical year. Additionally, MHWs have been implicated in the occurrence and persistence of thermal stratification, leading to changes in vertical mixing and nutrient availability in the water column (Chen et al., 2022). Such alterations in thermal stratification can have profound implications for the functioning of marine ecosystems and their resilience to climate change (Herring et al., 2015). The recently published Copernicus Ocean State Report 5 (Wakelin et al., 2021) documented the identification of extreme temperature events and their potential impacts

on important fish and shellfish stocks. However, a lack of systematic studies to elucidate the long-term relationship between the vertical stratification and MHWs hampers our understanding of the impacts of extreme temperature events on ecosystem stability.

Given the potential consequences of increased MHW frequency on the NWES region, it is essential to address the research

question: Is the NWES becoming more stratified due to the increased frequency of marine heatwave occurrence? Addressing this question requires a comprehensive assessment of long-term observational data, climate model simulations, and advanced analytical techniques to examine the relationships between MHW events, thermal stratification, and their ecological implications. This research will provide valuable insights into the potential impacts of MHWs on the NWES ecosystem and contribute to our understanding of the broader effects of climate change on marine environments.




## 2 Material and Methods

The three-dimensional water temperature and salinity data from Copernicus Marine Environment Monitoring Service (CMEMS) ocean physics reanalysis data (Table 1, prdoct ref. 1 & 2) is applied in this study. These data cover the NWES with assimilation model at 7 km horizontal resolution for the period 1993 to 2021 and at 1.5 km horizontal resolution for 2022. More details of the CMEMS products are given in Table 1.

Moreover, the CMEMS SST reanalysis is extended by the European Space Agency Sea Surface Temperature Climate Change Initiative (ESA SST CCI, Table 1 product ref. 1) Level 3 products for the period 1982-1992 (Merchant et al. 2019). This product has a spatial resolution of 0.05° by 0.05° for the Atlantic North-West Shelf region. All SST data are interpolated on the same spatial grid as the CMEMS product ref. 1 (Table 1), such that the 40-years period provides the baseline climatology reference period for computing the seasonally varying 90th percentile threshold defined as in Hobday et al. (2016). The matlab toolbox by Zhao and Marin (2019) is applied for detecting MHW events and to properly computing means and trends of MHW properties.

The potential energy anomaly is used as a measure of the degree of density stratification (Simpson, 1981):

$$\emptyset = \frac{1}{D}\int_{-H}^{\mu} gz(\bar{\rho} - \rho)\, dz, \tag{1}$$

in which

$$\bar{\rho} = \frac{1}{D}\int_{-H}^{\mu} \rho\, dz \tag{2}$$

is the vertical mean water density, and g = 9.8 m s$^{-2}$ is the gravitational acceleration. The instantaneous total water depth is given by D = η + H, with η and H being the sea surface elevation and the time mean water depth, respectively. The potential energy anomaly measures the amount of mechanical energy (per m$^3$) required to instantaneously homogenize the water column with given density stratification. The water density ρ was calculated (at 1 atm) following Millero and Poisso (1981):

$$\rho(S, T) = \rho_r + AS + BS^{1.5} + CS^2. \tag{3}$$

In eq. (3), S is the salinity of seawater in ppt (parts per thousand by volume). The reference density $\rho_r$, the coefficients A, B and C are also functions of temperature T in °C with expressions given by Millero and Poisso (1981):

$$\rho_r = 999.842594 + 6.793952 \times 10^{-2}T - 9.095290 \times 10^{-3}T^2 + 1.001685 \times 10^{-4}T^3 - 1.120083 \times 10^{-6}T^4$$
$$+ 6.536332 \times 10^{-9}T^5;$$



$A = 8.24493 \times 10^{-1} - 4.0899 \times 10^{-3}T + 7.6438 \times 10^{-5}T^2 - 8.2467 \times 10^{-7}T^3 + 5.3875 \times 10^{-9}T^4;$

$B = -5.72466 \times 10^{-3} + 1.0227 \times 10^{-4}T - 1.6546 \times 10^{-6}T^2;$

$C = 4.8314 \times 10^{-4}.$

Here S and T, as well as the sea surface elevation η and water depth H (see eq.1 and 2) are obtained from the CMEMS products.

Furthermore, the sensitivity of density stratification to the occurrence of MHWs is quantified with the ratio between the varying

heatwave days and varying water-stratified days (Chen et al., 2022):

$$r = \frac{\sum_i^n |N_i - \overline{N_n}|}{\sum_i^n |M_i - \overline{M_n}|}. \tag{4}$$

Here, the number of MHW days ($M$) and the number of days that the water column was stratified ($N$) are counted in each year

(i). The parameter n = 30 (i.e., 1993-2022) indicate the length of the computing period. The overline ($^-$) denotes the multiyear

mean.

**3 Results**

A MHW in each year is characterized by the number of events, the duration and the intensity of each event. As an example,

Figure 1b illustrates the detection of MHW events near the Dogger Bank region (Region 1 in Figure 1a), in the middle of the

North Sea in 2022. The MHW occurs multiple times throughout the whole year, even during winter. The duration of each

event varies from 5 days (e.g., event 4, 5) to 40 days (event 6, 7). The intensity, which measures the deviation of SST from the

threshold, reaches its maximum of 2 ℃ in late September.

MHWs are exceptionally active in the year 2022. The total number of days of MHWs reaches 140 days on average (Figure

1c), with the maximum exceeding 200 days in the English Channel. The lowest MHW days, 60 days, is found in the Norwegian

Trench. Throughout this year, the Celtic Sea experienced more than 7 MHW events. Apart from the events in 2022, MHWs

were also active in the years 1995, 1997, 2003, 2007, and 2014, during which 3 or more MHW events were observed (Figure

1d).

The occurrence of MHWs undergoes large temporal variations between the years 1993 and 2022 at periods of approximately

7 to 10 years, when MHW days appear much longer than during other years. During 1993-2022, the general duration of a

MHW event is around 10 to 20 days (Figure 1e). Longer durations are also observed. For example, in 2007, a duration with

more than 40 days duration was observed in the English and the Celtic Sea (Region 8, Figure 1a). The annual mean intensity

of MHWs ranges on average between 1 ℃ and 2 ℃, and shows insignificant differences with the occurrences of MHW, the

number of days or the length of duration (Figure 1f). However, regional dependence can be observed. For example, in the

Shetland-Irish Shelf (Region 6, Figure 1a), the mean intensity varies between 1 ℃ and 1.5 ℃ over the period 1993 to 2022. It

also becomes stronger in the Irish Sea (Region 7, Figure 1a) and the North Sea (Region 1~3, Figure 1a) over these periods. In



general, the annual mean intensity of MHWs occurring in the Norwegian Trench (Region 4, Figure 1a) can reach values up to 2 ºC to 3 ºC (Figure 1f), which is more intensive than in the other regions.

The mean and trend of MHWs over the past 30 years are shown in Figure 2. The lowest frequency of the occurrence of MHWs
is found at the southern North Sea and the English Channel, where on average only 1 ~ 2 MHW event occurred every year (Figure 2a). The Shetland-Irish Shelf and the Celtic Sea experienced 2 to 3 MHW events. Concurrently, all these regions had longer MHW periods (approximately 40 ~ 50 days) than the middle and northern North Sea (Figure 2b). Furthermore, it is found that MHW events last longer further to the south. The longest duration, 30 ~40 days, occurred near the coast of the North Sea and the English Channel (Figure 2c). This indicates that the MHWs that appear in the southern part of the NWES are
mostly continuous and long-term, while the MHWs that appear in the northern part of the NWES and the shelf edge are mostly intermittent and short-term.

The intensity of MHW shows a different geographical distribution than duration (Figure 2d). In the west part of the NWES, the 30 years mean intensity of MHW is approximately 0.5 ºC to 1 ºC. The lowest mean intensity is observed in the Irish Sea
and the English Channel. The MHW intensifies towards the east coast of the NWES. Along the coast of Denmark and Norway, the mean intensity reaches approximately 2.5 ºC ~ 3 ºC.

Over the past three decades, the frequency of MHW increased at a rate of 0.1-0.15 per year (Figure 2e). Correspondingly, the number of days experiencing MHWs increased by 1 to 4 days per year (Figure 2f). Coastal areas are where the number of days
increases the fastest, generally more than 2 days/yr over the study period. The fastest increasing region is the English Channel, while reaching or even exceeding 4 days/yr. The duration of the MHW events shows no significant trend in the NWES, except for the English Channel, where each MHW event was 1 to 2 days longer, corresponding to increasing MHW days (Figure 2g). The mean intensity (Figure 2h) shows a completely different trend in the North Sea and the rest regions. In the North Sea, MHWs tend to be less intense, whereas in the Shetland-Irish Shelf and the Celtic Sea, MHWs intensified at a rate of 0.02
~0.04 ºC/yr. In the English Channel and the west part of the southern North Sea, the annual mean intensity remained the same from 1993 to 2023.

The degree of density stratification, quantified by the potential energy anomaly (PEA) ø, is shown in Figure 3. Following Chen et al. (2022), the water column is considered stratified when ø≥50 J m$^{-3}$. The NWES is weakly stratified with ø≈70 J m$^{-3}$.
During summer, higher SST enhances the density stratification, leading to ø in summer approximately twice as high as the annual mean. The PEA is low in the southern North Sea, the Irish Sea, where the depth is small. Due to strong tides, the water column is generally well mixed in the English Channel (Pohlmann, 1996). In the middle North Sea, the Shetland-Irish Shelf and the Celtic Sea, the density stratification presents an obvious seasonal summer stratification (Figure 3). Over the whole year, the potential energy is around 50 J m$^{-3}$, while during the summer period it reaches 80 ~ 100 J m$^{-3}$. The northern North





Sea shows a seasonal cycle similar to that of the middle North Sea, but with a larger ø, both in summer period and in annual mean, due to larger depth. The potential energy anomaly in the Norwegian Trench is ~ 400 J m⁻³. This is much larger than what is observed for the NWES region as a whole.

The density stratification has exhibited a trend opposite to that of MHWs over the past 30 years. Figure 4 illustrates the 10-
year average PEA across three decades, spanning from 1993 to 2022. Evident shifts in stratification are discernible. An obvious transformation of stratification is observed. Between 1993 and 2022, a decreased at a rate of -1 to -2 J m⁻³/yr only in the eastern part of the southern and middle North Sea, as well as in the Celtic Sea. Conversely, in the eastern part of the middle North Sea, the northern North Sea and the Shetland Shelf, ø increased by approximately 1~1.5 J m⁻³/yr. During the second decade, the region where stratification grows, was greatly reduced. Only parts of the northern North Sea near the Norwegian Trench and
some areas of the Irish Shelf still maintained a growth rate of 1 to 2 J m⁻³/yr.  In the third decade, the entire North Sea tended to be less stratified. Especially in the middle and northern North Sea, where the potential energy anomaly decreased by -2 to -3 J m⁻³/yr. The transformation in the Norwegian Trench is consistent with that on the NWES, i.e., from an increased stratification from 1993 to 2012 to a decreased stratification in the most recent decade,  albeit at a relatively higher rate (6~ 8 J m⁻³/yr).


The PEA trend is further decomposed to that due to seawater temperature (T only, Figure 4 second row) and salinity (S only, Figure 4 third row). The former is related to the meteorological conditions while the latter is related to the regional salt and freshwater changes. A positive PEA trend due to T or S implies a reduced vertical gradient in temperature or salinity, respectively. The decomposition reveals that in the North Sea region, both the water temperature and the salinity cause the
weakening of stratification. The changes in trend in the Norwegian Trench is the result from the changing salinity trend (via freshwater inflow  variability of the Baltic) (Tinker et al., 2016).

The sensitivity of stratification to the occurrence of marine heatwaves (MHWs), as quantified using equation 4, is demonstrated in Figure 5. The illustration unequivocally indicates a profound linkage between summer stratification and MHWs, i.e., the
seawater temperature T, particularly from June to September, within the southern North Sea expanse, notably in the eastern sector extending up to a longitude of 4.5°E (German Bight). This observation is consistent with the conclusions drawn by Chen et al. (2022). Another geographic area exhibiting a significant correlation between the presence of summer stratification and MHW events is the Shetland Irish Shelf, which corresponds to Region 6 in Figure 1a. However, upon extending the temporal analysis window to encompass the entire year (i.e., all 12 months), it becomes evident that the northern North Sea (Region 3,
Figure 1a), the Celtic Sea (Region 8, Figure 1a), and the Norwegian Trench (Region 5, Figure 1a) also evince correlations with MHWs. This suggests that the manifestation of MHWs, especially those occurring during the winter season (see, e.g., Figure 1b), instigate temperature disparities between the sea's surface and its deeper layers, consequently giving rise to thermal stratification. Consequently, the escalating trend in MHW occurrences exerts an influence not only on the density stratification





within the southern North Sea but also across other locales within the NWES, primarily due to the mounting incidence of
MHWs during winter periods. However, in the English Channel (Region 4, Figure 1a) and the western portion of the southern
North Sea (Region 31, Figure 1a) lying westward to the 4.5°E longitude, the water experiences annual thorough mixing
attributed to tidal forces, thereby mitigating the impact of MHWs on density stratification.

**4 Discussion**

MHW events have become more frequent and prolonged over the period 1993-2022. The total number of days experiencing
MHWs showed an upward trend, particularly in the English Channel with the rate of 2~4 days/yr. These findings align with
previous studies, which reported increasing MHW occurrences globally (e.g., Oliver et al. (2018) and Smale et al. (2019)) and
in coastal regions (Marin et al., 2021). During the past 30 years, the most prolonged MHW found in the nearshore areas is
mainly attributed to the long-term changes in mean SST, which increased fastest at the coastal oceans (Marin et al., 2021).
Moreover, we observed an increase in MHW duration, indicating the potential for prolonged impacts on marine ecosystems
(Frölicher and Laufkötter, 2018; Suryan et al., 2021). However, the results also show the increase in MHW frequency and
length may unnecessarily coincide with the intensification of SST. In the middle North Sea and near the Danish coast, the
trend of mean MHW intensity is -2~-6 J m$^{-3}$/yr, which appears contradictory to the increase of MHW frequency and duration.

One possible explanation for the decrease in mean MHW intensity in the North Sea is the influence of large-scale climate
patterns, such as atmospheric circulation changes. Woollings et al. (2018) has demonstrated a weakening of the North Atlantic
Jet Stream and an increase in atmospheric blocking events over the North Atlantic region. These changes can lead to stagnant
atmospheric conditions and the trapping of warm air masses over the NWES, resulting in prolonged periods of high SST and
MHW events. The increase in MHW frequency and duration may be a consequence of these altered atmospheric circulation
patterns rather than a direct result of SST intensification. Furthermore, internal variability in the regional ocean, especially
local processes that affect SST play an important role (Marin et al., 2021). This includes variations in the fresh water/salt
exchange and stratification (Mathis et al., 2015). Changes in these processes, particularly the Baltic Sea inflow, can affect the
stability of the water column and result in a localized decrease in mean MHW intensity, while outside the North Sea region,
the intensity of MHWs increases due to different oceanic processes and heat transport mechanisms in the North Atlantic Ocean
(Plecha et al., 2021). In addition, it is important to note that the observed decrease in mean MHW intensity in certain areas
does not negate the overall increase in MHW frequency and duration. Climate change, with its warming trends and interactions
with atmospheric and oceanic conditions, is a key driver of the intensification and increased occurrence of extreme events like
MHWs. While localized decreases in mean intensity may be present, the number and duration of MHW events are surpassing
these reductions. These findings highlight the complex interactions between climate change, atmospheric circulation patterns,
and regional oceanic processes in shaping MHW characteristics.




Despite the more frequent and prolonged MHW leading to a more stable water column, the potential energy anomaly (ø), a measure of stratification, showed a decreasing trend in the North Sea. This suggests that the MHW events and density stratification in the NWES region are not directly related. One notable evidence is that the region where mean MHW intensity shows a downward trend overlaps with the area of reducing density stratification.


In general, MHW events lead to a significant increase in SST, resulting in intensified thermal stratification during these events (Chen et al., 2022). The analysis of annual mean SST trends in the NWES over the past three decades reveals a positive trend with a rate of change of 0.03~0.05 ºC/yr (EU Copernicus Marine Service Product, 2022b). Interestingly, the PEA due to seawater temperatures shows an opposite trend, particularly from 2003 to 2022 (Figure 4). This suggests that the temperature

gradient in the water column is decreasing despite the warming climate, primarily driven by strong winter warming and resulting in a weakening of the thermal stratification (Mathis and Pohlmann, 2014). This phenomenon is evident in the northern North Sea and the Celtic Sea, where the emergence of thermal stratification exhibits strong correlations with the occurrence of MHWs during winter seasons (Figure 5). Therefore, the rise in SST caused by increased MHW events is insufficient to counterbalance the overall weakening of thermal stratification due to seawater warming. Additionally, other factors and

processes, such as oceanic circulation patterns and mixing mechanisms, may contribute to the observed changes in stratification (Guihou, et al., 2017). Increased river runoff can lead to stronger salinity decreases at the sea surface compared to deeper layers, intensifying the stratification in terms of an increasing vertical salinity gradient (Lehmann, et al., 2022). This can be seen at the estuarine zones, exemplified by the German Bight in the southern North Sea (Chegini, et al., 2020). The analysis of PEA trends due to salinity also highlights the significant impact of variation of the Baltic discharge on stratification in the

Norwegian Trench of the North Sea  (Tinker et al., 2016).

It is notable that, within the NWES, the variability in salinity exerts a more pronounced influence on stratification compared to temperature variability. However, it is essential to recognize that climatic factors also potentially impact salinity fluctuations in the NWES. Schrum et al. (2016) reported a freshening trend in the North Sea attributed to increased river runoff and Baltic

discharge, both intricately linked to an intensified water cycle and amplified net precipitation in mid to high-latitude regions (Collins et al., 2013; Levang and Schmitt, 2015). Moreover, atmospheric heatwave events, known as significant contributors to MHW occurrences (Hobday et al., 2016), can notably affect precipitation and evaporation dynamics (Miralles et al., 2019), thereby further influencing river runoff and Baltic discharge (Lehmann et al., 2022). As a prospective avenue of research, delving into the NWES and Baltic as an integrated system would yield intriguing insights. Exploring the interplay between

European continent-wide precipitation, evaporation patterns, and the evolving stratification trend in the NWES presents an intriguing prospect. Gaining comprehension of these intricate interactions would offer valuable revelations about the multifaceted mechanisms steering shifts in stratification and their intricate connections to regional climate dynamics.



## 5 Conclusion

Leveraging the wealth of high-resolution data furnished by the Copernicus Marine Environment Monitoring Service, we have
analyzed the occurrence of MHW events and their underlying characteristics spanning the last three decades within the NWES
region.

Our analysis revealed multiple MHW events throughout the year, including during the winter season. We find that despite
showing spatial variations, MHW has tended to become more frequent and prolonged over the past three decades in the NWES.
The temporal dynamics of MHWs reveal a noteworthy trajectory, augmenting at a rate of 0.1 to 0.15 events per year on average,
contributing to a rise in annual occurrence by a range of 1 to 4 days. It is evident that coastal areas are the epicenters of this
phenomenon, experiencing the swiftest augmentation in MHW duration when juxtaposed against other regions within the
NWES. However, the NWES did not show a trend toward stronger stratification due to MHW occurring more frequently and
lasting longer. On the contrary, it becomes less stratified, especially the middle and northern North Sea region.


A closer examination of seawater temperature trends reveals that the rise in SST caused by increased MHW events is
insufficient to offset the overall weakening of thermal stratification due to seawater warming. This is evident in the northern
North Sea, where the emergence of thermal stratification exhibits a strong correlation with the occurrence of MHWs during
winter seasons. However, the intricate dynamics extend beyond temperature alone. The variance of salinity has a significant
impact on the trend of change in density stratification. In particular, the influence of Baltic discharge, a veritable fulcrum of
internal variability, emerges as a paramount process dictating the trajectory of changes in density stratification within the North
Sea. The intricate interplay of freshwater influx from the Baltic Sea, influenced by climatic factors such as intensified water
cycles and augmented net precipitation, intricately shapes the spatial distribution of salinity patterns within the North Sea
realm. It is imperative to treat the NWES and Baltic as integral components of a larger, interconnected system. The
interdependency between these domains necessitates a comprehensive approach, one that transcends arbitrary boundaries and
delves into the subtle threads linking various climatic, oceanographic, and hydrological factors.

In summary, the interplay between temperature, salinity, and their intricate interactions with MHWs, Baltic discharge, and
broader climatic phenomena collectively weave the intricate tapestry of density stratification trends within the North Sea
region. This multifaceted narrative underscores the necessity of adopting a unified perspective, one that considers the complex
interdependencies and feedback loops that characterize the intricate dance of nature's forces in this vital marine expanse.

### Data availability

The datasets presented in this study can be found in the CMEMS online repository. Details are listed in Table 1 of this paper.



**Author contribution**

WC conceptualized the study, analyzed data, and wrote this article. JS contributed to the writing of the article and quality control.

**Competing interests**

The authors declare that they have no conflict of interest.

**Acknowledgement**

This study was funded by the EU Green Deal project REST-COAST: Large scale restoration of coastal ecosystems through rivers to sea connectivity (grant agreement 101037097). Joanna Staneva acknowledges OLAMUR project: Offshore Low-trophic Aquaculture in Multi-Use Scenario Realisation (grant agreement 101094065). We also thank Lorena Moreira Mendez and Karina von Schuckmann for their comments and great help in improving the quality of this paper.

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





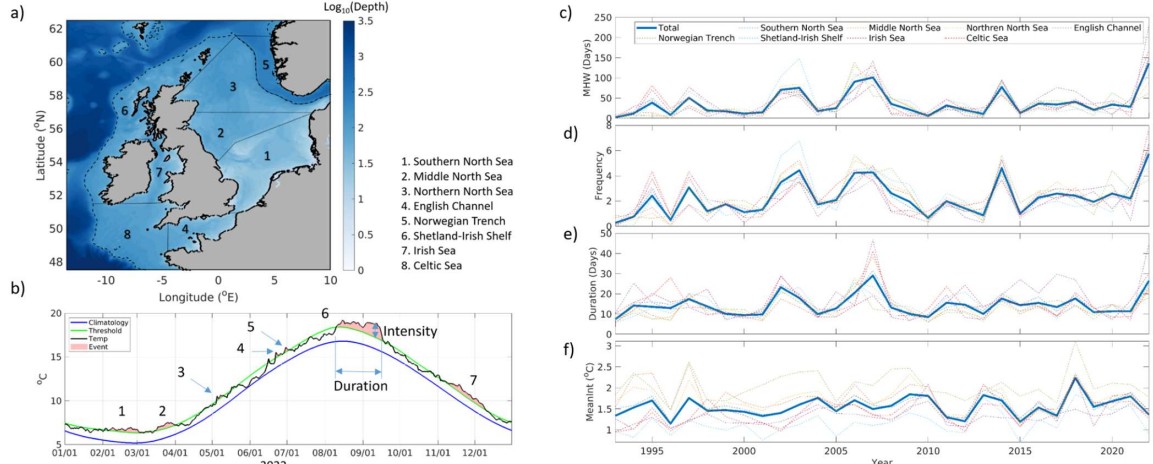

**Figure 1: a) Map of North West European Shelf Sea with sub-region division (data from Table 1 ref. 1). Dashed curve indicates 200 m isobath. b) Detection of MHW events and their characteristics in 2022 (data from Table 1 ref. 1 & 2); c)-f) Variations of MHW characteristics between 1993 and 2022, with the bold solid curve indicating the mean of total subdomains of the NWES (daily SST data from Table 1 ref. 1 & 2).**


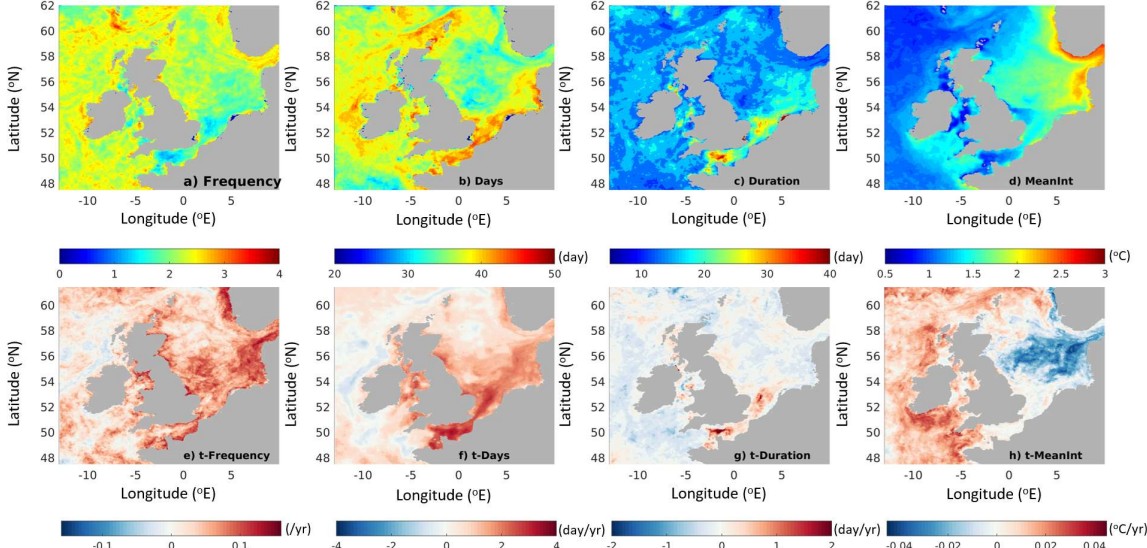

**Figure 2. Mean and trend of MHW in the past 30 years (1993-2022). Dashed lines indicate 200 m isobath. (daily SST data from Table 1 ref. 1 & 2).**




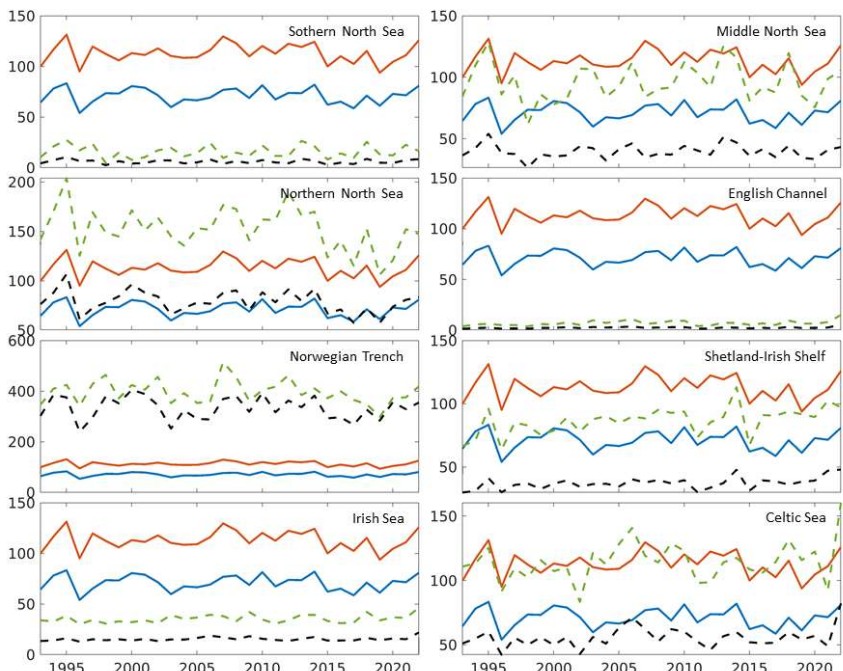

**Figure 3. Potential energy anomaly (ø, J m-3) between 1993 and 2022. The solid curves denote the summer (June-July-August) mean (red solid) and annual mean (blue solid) over the spatial mean of the entire NWES domain, respectively. The green and black dashed curves are similar as the red and blue curves, but for different subdomains (see Figure 1a).**


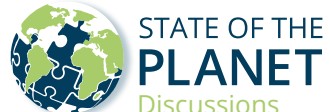

**Figure 4. Trend of potential energy anomaly (ø, J m-3 yr-1) in the last 30 years. The first row (total) shows ø computed with eq. 1, where density depends on both temperature (T) and salinity (S) in the water column. The second and third row are similar as the first row but with density depends only on either T or S, respectively.**






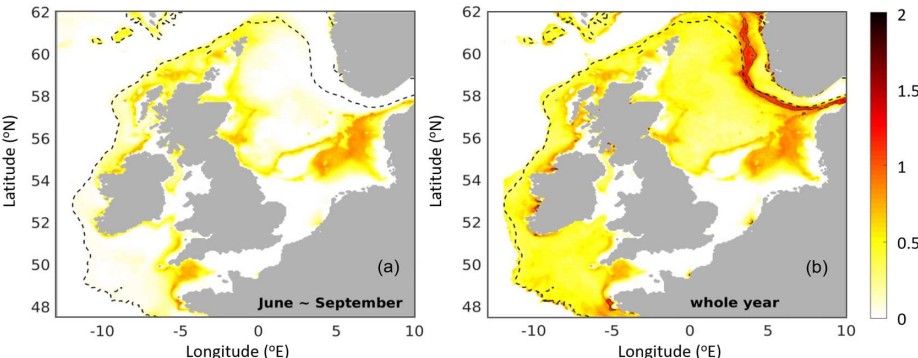

**Figure 5. Ratio of the number of water stratification days to the number of MHW days for (a) June to September (summer period)**
**and the whole year. The ratio is computed with Eq.4 using multi-year water temperature, salinity at different depths for 1992 to**
**2022 (Details are in Table 1 ref 1 & 2). The thin dashed line indicates the 200 m isobaths.**

**Table 1. CMEMS products used in this study**

| Product ref. no. | Product ID & type | Data access | Documentation |
|---|---|---|---|
| 1 | NWSHELF_MULTI YEAR_PHY_004_00 9, Numerical models | EU Copernicus Marine Service Product (2021) | Quality information Document (QUID): Renshaw et al. (2021) <br> Product User Manual (PUM): Tonani et al. (2022a) |
| 2 | NORTHWESTSHEL F_ANALYSIS_FORE CAST_PHY_004_013 , Numerical models | EU Copernicus Marine Service Product (2022a) | QUID: Tonani et al. (2022b) <br> PUM: Tonani et al. (2022c) |
