# Peer review of "Characteristics and Trends of Marine Heatwaves in the Northwest European Shelf and the Impacts on Density Stratification Wei Chen1, Joanna Staneva1"

_State of the Planet, 2023_

## Author Comment (AC1)

**Reply to Reviewer 1**

Dear Reviewer,

We want to thank you for your dedicated time to review our manuscript. Your input has helped improve the clarity and robustness of the document. The changes are marked in the manuscript, as well as the reply to your comments below (in blue).

This study uses a combination of satellite observations and ocean reanalysis data to research the characteristics and trends of marine heatwaves on the northwest European Shelf. Spatial variations of the marine heatwave metrics and their trends are emphasized. The author further looked into the relationship between the marine heatwaves and water column stratification on the shelf and they have found a decreasing trend of the stratification despite the increasing surface marine heatwaves. It is an interesting study, however, some further analyses are necessary to support the conclusion from the study, so I would suggest major revisions before a publication can be considered.

Below, we present a point-to-point reply to all comments.

The authors use two datasets but there is a lack of cross-validation of the two. Also, for the ocean model data, some validation for the research region is necessary.

The two datasets mentioned by the reviewer are the CMEMS reanalysis data, covering the period 1993-2022, and the ESA CCI SST data, spanning the years 1982-1992. In the manuscript we provided references to them. As stated in both the 'quality information document' and the 'user manual' of the CMEMS product (refer to Table 1 in the manuscript), the latter dataset, specifically the ESA CCI SST L3 satellite observations, is utilized for assimilating ocean model data and generating the L4 CMEMS reanalysis data, which constitutes the former dataset. Regarding the validation of the model's temperature and salinity, the data are compared with in-situ observations from the World Ocean Database, mooring data, and the multi-model ensemble of multi-year products, an internal CMEMS product (see the 'quality information document', Table 1). In this reply, we provided references to them:

QUID: https://catalogue.marine.copernicus.eu/documents/QUID/CMEMS-NWS-QUID-004-009.pdf

PUM: https://catalogue.marine.copernicus.eu/documents/PUM/CMEMS-NWS-PUM-004-009-011.pdf

In the revised manuscript, we have incorporated the above information in the first and second paragraph of section 2 'Material and Methods' as follows: "The modelled temperature and salinity are validated through comparisons with in-situ observations from the World Ocean Database, mooring data, and the multi-model ensemble of multi-year products, an internal CMEMS product.", and "The ESA dataset is also employed as

observational data for assimilating CMEMS data (see QUID and PUM of the product, Table 1).

The authors propose some subsurface warming causes the opposite trends between marine heatwaves and stratification. However, the subsurface data is now shown in the study, which is necessary to confirm the conclusion.

*In our study, we did not present subsurface temperature data; instead, we showcased the Potential Energy Anomaly (PEA) resulting from temperature alone (Figure 5). This was computed based on temperatures at various layers of the water column, serving as a parameter to quantify the temperature heterogeneity of the water column. Typically, temperatures in deep water columns are lower than those at the surface. The reduction in PEA indicates a diminishing temperature difference between the surface and the subsurface. As stated in the discussion, the annual mean Sea Surface Temperature (SST) trends in the NWES over the past three decades have shown an increase. Consequently, the declining trend in PEA, attributed solely to temperature, can be linked to the warming of the subsurface water. This subsurface warming is predominantly influenced by robust winter warming (Mathis and Pohlmann, 2014). The lower water column retains the memory of winter warming for a more extended period compared to the surface (Chen et al., 2022).*

*Reference*

*Mathis, M., & Pohlmann, T.: Projection of physical conditions in the North Sea for the 21st century, Clim. Res., 61, 1-17, https://doi.org/10.3354/cr01232, 2014.*

*Chen, W., Staneva, J., Grayek, S., Schulz-Stellenfleth, J., & Greinert, J.: The role of heat wave events in the occurrence and persistence of thermal stratification in the southern North Sea, Nat. Hazards Earth Syst. Sci., 22, 1683-1698, https://doi.org/10.5194/nhess-22-1683-2022, 2022.*

To clarify this point, we revised the discussion text as follows: "As this parameter quantifies the temperature heterogeneity of the water column, the decrease in PEA suggests a reduction in the temperature difference between the surface and the subsurface. With the observed increasing trend of NWES SST in response to a warming climate, the decline in PEA due to temperature can be solely attributed to the warming of the subsurface water. This warming is primarily driven by strong winter warming, leading to a weakening of thermal stratification (Mathis and Pohlmann, 2014). Additionally, the lower water column retains the memory of winter warming for a longer duration compared to the surface (Chen et al., 2022).

In lines 118-119, the authors suggest that there is a periodicity of 7-10 years in regard to the MHWs occurrences. This is not obvious in Fig. 1. There needs to be more statistical analysis on that statement.

Thank you. We removed this statement in the revised manuscript.

Line 134-136: This is an interesting finding. Is there a climate driver or dynamic process that drives this spatial variation?

We are thankful to the reviewer for initiating this insightful discussion. Following this sentence, we extended the subsequent paragraph with additional content: "The occurrence of Marine Heatwaves (MHWs) can be primarily attributed to two drivers: local air-sea heat exchange resulting from abnormally high air temperatures and nonlocal heat transport via ocean advection (Gupta et al., 2020; Schlegel et al., 2021). The atmospheric factor emerges as the predominant driver of MHWs in the southern to middle North Sea (Chen et al., 2022; Mohamed et al., 2023). Nonlocal heat fluxes, such as the influx of warm Atlantic water into NWES, may be responsible for the development of MHWs.

Compared to the long-term average, higher seawater temperatures will result in more heat fluxes into the NWES by the North Atlantic shelf current, particularly in the English Channel and the Shetland-Irish Shelf (zones 4 and 6 of Figure 1a). The heightened seawater makes these areas more prone to experiencing MHW compared to regions less affected by the North Atlantic current, such as the Norwegian Trench (zone 5, Fig. 1a). This may explain why these regions have more days with MHW. Furthermore, the mean intensity of MHW in these two regions is notably lower than in the Norwegian Trench (Figure 2d), supporting the assertion. The lowest mean intensity is observed in the Irish Sea and the English Channel. The MHW intensifies towards the east coast of the NWES. Along the coast of Denmark and Norway, the mean intensity reaches approximately 2.5 $^o$C to 3 $^o$C. However, compared to the southern NWES, the shorter durations and higher frequencies of MHW in its northern region may be attributed to the distinct characteristics of climate drivers in their respective areas. This is because atmospheric influences, in contrast to oceanic influences, exhibit larger variability in affecting SST (Tinker and Howes, 2020). Other drivers, such as local wind (Mohamed et al., 2023), may introduce further uncertainties to the occurrence and persistence of MHWs. Identifying the dominant drivers of MHW features in NWES requires a systematic investigation of the relationship between air and sea temperatures in various regions. However, this detailed analysis is not elaborated in this paper due to space constraints.

*References*

*Chen, W., Staneva, J., Grayek, S., Schulz-Stellenfleth, J., & Greinert, J.: The role of heat wave events in the occurrence and persistence of thermal stratification in the southern North Sea, Nat. Hazards Earth Syst. Sci., 22, 1683-1698, https://doi.org/10.5194/nhess-22-1683-2022, 2022.*

*Gupta A. S., Thomsen M., Benthuysen J. A., Hobday A. J., Oliver E., Alexander L. V., et al.: Drivers and impacts of the most extreme marine heatwaves events. Sci. Rep. 10, 19359. http://doi.org/10.1038/s41598-020-75445-3, 2020.*

*Mohamed, B., Barth, A. and Alvera-Azcarate, A.: Extreme marine heatwaves and cold spells events in the Southern North Sea: classification, patterns, and trends, Front. Mar. Sci., 19, https://doi.org/10.3389/fmars.2023.1258117, 2023.*

*Schlegel R. W., Oliver E. C. J., Chen K.: Drivers of marine heatwaves in the northwest atlantic: the role of air–sea interaction during onset and decline. Front. Mar. Sci. 8. http://doi.org/10.3389/FMARS.2021.627970/BIBTEX, 2021.*

*Tinker J., Howes E. L.: The impacts of climate change on temperature (air and sea), relevant to the coastal and marine environment around the UK, MCCIP Science Review. http://doi.org/10.14465/2020.arc01.tem, 2020.*

line 158: Are all shelf regions present summer stratification?

Clear summer stratification is observed in the Shetland-Irish shelf region, as presented in Figure 3, with PEA (ø) ≥ 50 J m$^{-3}$ during summer and ø < 50 J m$^{-3}$ for the annual mean. The Celtic Sea area also exhibits summer stratification with a seasonal cycle, although not as pronounced as the Shetland-Irish Shelf. Its annual mean PEA hovers around 50 J m$^{-3}$. This difference is attributed to the southern location of the Celtic Sea, resulting in a longer warming period compared to the northern regions and stratified shelf waters during autumn. In the revision, we rephrase this sentence as follows: 'In the middle North Sea and the Shetland-Irish Shelf, ø≥50 J m$^{-3}$ during summer and ø<50 J m$^{-3}$ for the annual mean, indicating clear seasonal summer stratification (Figure 3). The Celtic Sea also exhibits seasonal cycles in PEA, with ø ranging from 110~120 J m$^{-3}$ during the summer and around 50 J m$^{-3}$ over the entire year. The large annual mean PEA is mainly attributed to the extended warming period and stratification during autumn.

The Norwegian Trench is not shown in Fig. 1.

The Norwegian Trench is labeled with number 5 in Fig.1a).

Line 206 and 214: what is "SST intensification"?

We revise it to SST heightening.

Line 238-239: a scatter plot may show the relationship better.

We appreciate the reviewer's suggestion. This statement is concluded based on the comparison of the increasing SST and the decreasing PEA due to temperature (Figure 4 middle rows). We believe that this statement becomes clear in the revised manuscript after we rewrite lines 256~264. Nonetheless, using the Celtic Sea as an example, we present a scatter plot in this reply to illustrate the relationship between the rise in SST caused by Marine Heatwaves (MHW) (the mean intensity of MHW) and the PEA due to temperature for the periods 1993-2022. There is no obvious linear correlation between the two variables (with $R^2$=0.1). However, it is found that the PEAs in more recent years are lower than those in earlier periods. Moreover, the mean intensity in more recent years (1.6~1.8°C) is larger than in earlier periods (1.2~1.4°C). These are the same features one can observe from Figure 2h and Figure 4 (middle row) of the manuscript, which provides additional spatial distribution of the variables of Figure R1. Considering Figure R1 does not add much new insight regarding the relationship, and the restrictions of the figure numbers required by the Ocean State report, we decide not to include this figure in the updated manuscript.

[Figure]

Figure R1. Scatter plot of the annual mean intensity due to MHW and annual mean PEA due to seawater temperatures at different years (in color scatter points).

---

## Author Comment (AC2)

**Reply to Reviewer 2**

Dear Reviewer,

We want to thank you for your dedicated time to review our manuscript. Your input has helped improve the clarity and robustness of the document. The changes are marked in the manuscript, as well as the reply to your comments below (in blue) .

Review of Is the North West European Shelf becoming more stratified with the occurrence of marine heatwaves? by Wei Chen and Joanna Staneva

This works looks at an interesting process: the possible changes in stratification of the water column that may occur with the increasing frequency and intensity of marine heat waves. As such the works lacks to settle which temporal scales the authors are working with. Is the average stratification over the year, or the stratification surrounding a MHW event? It is not clear how long the enhanced stratification caused by a MHW event would last in a region subject to strong tidal currents. The authors find that indeed, stratification appears to be weakening, which may appear counter-intuitive but I think it stems from the temporal scale mentioned above.

Thank you for the comment. As depicted in Figure 3, the stratification is averaged over years (blue solid lines) and over summer periods (red solid lines). Recognizing that merely describing it in the figure caption is insufficient, we have augmented the figure descriptions in the revised manuscript. In the revised 'Results' section, we have added: "In this study, only annual mean and summer period (June to September) mean stratification are considered."

I am not a huge fan of questions in titles, and in this case, in which the answer appears to be negative, I would suggest to rephrase the title so that it is more informative.

We express our gratitude to the reviewer for the suggestion. The title is revised to: "Characteristics and Trends of Marine Heatwaves in the Northwest European Shelf and the Impacts on Density Stratification"

I include here below a few comments that could maybe help in improving this work.

Thank you!

The main comment, as mentioned above, would be to establish from the beginning at which temporal scales do the authors think the MHWs would have an effect on stratification, and perform tests at different time scales to assess how long the effects of MHWs are felt in the water column. As the place is limited I would suggest to cut on the part of the MHW

description, as there is already previous work on that on this region (e.g. Mohamed et al 2023) and focus more on the stratification part.

Thank you for the suggestion. In the updated manuscript, we have incorporated the referenced study, which was primarily focuses on the southern North Sea and not published when we initially submitted our work. We would like to clarify here that our study encompasses a comparative analysis of Marine Heatwaves' (MHWs) characteristics and trends across the entire Northwest European Shelf (NWES) over recent decades, including the Southern North Sea. This comprehensive analysis involves examining and comparing the features and trends of MHWs in different regions and their correlation with the trend of density stratification. Our findings reveal continuous and long-term MHWs in the southern North Sea. In contrast, the northern part of the NWES, particularly the shelf edge zone, experiences mostly intermittent and short-term MHWs.

To the best of our knowledge, our study represents the first comprehensive investigation into the patterns and trends of MHWs across the entire NWES and their connections with long-term density stratification features.  Taking the southern North Sea as an example, a comparison between MHW patterns and trends (Figure 2) and stratification patterns and trends (Figure 4) shows no clear density trends, even as MHWs become longer and more frequent. Therefore, we believe that including the entire NWES and analyzing patterns and trends in different regions is essential for assessing MHW impacts on stratification trends across these diverse areas.

Detailed comments:

The Abstract contains a last part with too general wording (starting at "The outcomes of this research transcend theoretical confines...") that I don't think belong to an abstract (-> Conclusion?)

Thank you for this suggestion. We moved this part to the conclusion.

line 67: 1982 - 2022 I guess?

We update the texsts, which clarified this issue.

line 178: salinity is related to salt and freshwater discharges. But in figure 4, last row, there is no signal of the many rivers (Scheldt-Rhine-Meuse, Thames, Elbe?) that flow into the North Sea. How is this possible? It may be that the signal is too weak compared to the offshore signals (Norwegian coast, Atlantic Sea), so maybe it would be good to limit these figures to the North Sea Shelf. Otherwise it is only the offshore features that can be discussed.

Indeed, the reviewer is correct that the signal in the Norwegian and the Atlantic Sea is too strong compared to the shelf sea. We agree with the reviewer to mask these areas in the revision and minimize the range for contour plots. The updated plots (see below as well as in the revised manuscript) clearly demonstrate the impact of the rivers (e.g., the Rhine River and the Elbe River) on the density stratification.

[Figure]

Figure 4. Trend of potential energy anomaly (ø, J m$^{-3}$ yr-1) in the last 30 years. The first row (total) shows ø computed with eq. 1, where density depends on both temperature (T) and salinity (S) in the water column. The second and third row are similar as the first row but with density depends only on either T or S, respectively.

Figure 1. It is confusing that there are numbers in the panel a and b but they do not refer to the same things. The dotted line in panel a is difficult to see. Is the panel b a global assessment of MHWs in the whole domain? Not very informative as almost all signals are damped down by the averaging. Panels c to f: the individual lines are impossible to see.

In panel a, the black dotted line is replaced by white solid line. We change numbers in panel b to I, II, III, IV, …, to denote 7 MHW events. The subplot b illustrates the detection of MHW events near the Dogger Bank region (Region 1 in Figure 1a). We stated in the revised manuscript (First paragraph of section 3). We revised description in the caption for the updated manuscript. We further bolder individual lines in panels c to f.

[Figure]

Figure 1: a) Map of North West European Shelf Sea with sub-region division (data from Table 1 ref. 1). Dashed curve indicates 200 m isobath. b) Detection of MHW events and their characteristics in 2022 (data from Table 1 ref. 1 & 2) near the Dogger Bank region in the southern North Sea (region 1 in panel a); c)-f) Variations of MHW characteristics between 1993 and 2022, with the bold solid curve indicating the mean of total subdomains of the NWES (daily SST data from Table 1 ref. 1 & 2).

Figure 3. This should be a central figure of this paper, as it presents the evolution of stratification over time, which is what the title claims the paper is about. But it is just slightly mentioned in the text, and in fact as the data are presented I think it is difficult to extract meaningful information. The text says "During summer, higher SST enhances the density stratification, leading to ø in summer approximately twice as high as the annual mean" but the lines in figure 3 (all panels) look quite homogeneous and no intra-annual variations are observed. I think the authors could get rid of figure 2 (and refer to results in literature) and expand figure 3 in two figures, maybe doing a short-term analysis of the effects of MHWs in the stratification and another with longer-term trends (i.e. your figure 3) but which would present the data more clearly: add grid lines, maybe a line showing the average value, and explain better what the green and black lines are (the caption says "The green and black dashed curves are similar as the red and blue curves, but for different subdomains" which I do not understand).

We decided to retain Figure 2 because we haven't found literature MHW features and trends in the entire North West European Shelf (NWES) region over the last 30 years. The most recently published paper (Mohamed et al., 2023) only studied MHWs and cold spells in the southern North Sea, which may not be representative of the entire NWES. As we clarified in response to your previous comment, there is no clear trend in stratification in the southern North Sea. We believe Figures 2 and 4 are essential for comparing patterns and trends in different regions. Moreover, the temporal variations in Figure 1c~f and Figure 2 only depict the subdomain-averaged annual mean characteristics of MHWs and Potential Energy Anomaly (PEA). However, they lack spatial patterns.

We revised the caption of Figure 3 for better description: 'Figure 3. Potential Energy Anomaly (ø, J m$^{-3}$) between 1993 and 2022. The solid curves denote the spatial mean PEA of the entire NWES domain. The red curves represent the summer period (June-July-August), and the blue curves represent the annual mean. The spatial mean PEA of different subdomains of the NWES (see Figure 1a) is indicated by dashed curves, with green and black dashed curves for the annual mean and summer mean, respectively.'

Figure 5. As for figure 4, in figure 5 variations at the Norwegian trench overshadow variations over the shelf. Also, the colorbar goes up to 2 but I would say 1.5 would be better?

Thank you. We updated Figure 5 with colorbar goes up to 1.5.

[Figure]

Figure 5. Ratio of the number of water stratification days to the number of MHW days for (a) June to September (summer period) and the whole year. The ratio is computed with Eq.4 using multi-year water temperature, salinity at different depths for 1992 to 2022 (Details are in Table 1 ref 1). The thin dashed line indicates the 200 m isobaths.